

# A high-resolution inventory of air pollutant emissions from crop residue burning in China

Xiaohui Zhang[1], Yan Lu[1], Qin'geng Wang[*1, 2], Xin Qian[1, 2]

[1]State Key Laboratory of Pollution Control and Resources Reuse, School of the Environment, Nanjing University, Nanjing, 210023, China
[2]Jiangsu Collaborative Innovation Center of Atmospheric Environment and Equipment Technology (CICAEET), Nanjing University of Information Science & Technology, Nanjing, 210044, China

*Correspondence to*: Qin'geng Wang (wangqg@nju.edu.cn)

**Abstract.** Crop residue burning is an important source of air pollutants and strongly affects the regional air quality and global climate change. This study presents a detailed emission inventory of major air pollutants from crop residue burning for the year of 2014 in China. Activity data were investigated for 296 prefecture-level cities, and emissions were firstly estimated for each city and then redistributed using 1-km resolution land use data. Temporal variation was determined according to the farming practice in different regions. The MODIS fire product was applied to verify the spatial and temporal variations of the inventory. Results indicates that the total emissions of BC, OC, $PM_{2.5}$, $PM_{10}$, $SO_2$, $NO_X$, $NH_3$, $CH_4$, NMVOC, CO and $CO_2$ from crop residue burning (including open and household fuel burnings) were estimated to be 0.16, 0.82, 2.30, 2.66, 0.09, 0.70, 0.14, 0.81, 1.70, 13.70 and 309.04 Tg, respectively. Rice, wheat and corn were the three major contributors, but their relative contributions varied with region and season. High emissions were generally located in the eastern China, central China and northeastern China, and temporally peaking in June and October relating with harvesting time. The spatially and temporal distributions agree well with the fire pixel counts from MODIS. Uncertainties were estimated using the Monte Carlo method. This study provides a useful basis for air quality modeling and the policy making of pollution control strategies.

## 1 Introduction

The burning of crop residue is an important source of air pollutants, such as particulate matter (PM), sulfur dioxide ($SO_2$), nitrogen oxides ($NO_X$), carbon monoxide (CO), black carbon (BC) and volatile organic compounds (VOCs), which may have remarkable negative effects on climate change (Zhang et al., 2008), regional air quality (Cheng et al., 2014; He et al., 2015; Huang et al., 2013; Li et al., 2010) and human health (Gadde et al., 2009; Zhang et al., 2016a). As a large agricultural country, China is rich in crop residue resources, with most (85 %) being corn, wheat and rice straw (Cao et al., 2008a; Li et al., 2017a). In recent years, the many incentive and mandatory polices implemented by government have resulted in much straw being returned to the field or used as livestock feed and biofuel materials. Nevertheless, the proportion of straw being burned remains high, at 10 %−50 % in different regions of China. Because crop rotation is a popular farming practice in



China, crop residues are more often than not burned in fields before the next planting of crops, expect for those being returned to the soil or used for other purposes. Consequently, although pollutant emissions from crop residue burning through a year are less than those from other anthropogenic sources, the emission intensity may be relatively high in certain places and at certain times, because crop residue burning is unevenly distributed across China and through the year (Ni et al.,

2015; Shon, 2015). As a result, crop residue burning in China often plays an important role in episodes of haze pollution, especially during or after harvest seasons (Chen and Xie, 2014; Zhang et al., 2017).

A high-resolution inventory of pollutant emissions from crop residue burning, in terms of both spatial and temporal variations, is essential for studies of environmental effects. Many researchers have estimated pollutant emissions from crop residue burning in China. However, almost all existing studies focused only on provincial emissions across China, without

considering the detailed distribution within a province. Representative studies conducted in recent years include those of Huang et al. (2012), Sun et al. (2016), Wang and Zhang (2008), Zhang et al. (2008), Zhang et al. (2016b) and Zhou et al. (2017). Crop residue is usually burned in agricultural areas, which generally account for only a small fraction of the total administrative area. A spatial distribution with finer resolution based on more detailed data of agricultural activity and land use is therefore expected. Meanwhile, previous studies either did not consider the temporal variation (e.g., Chen et al., 2015;

Zhang et al., 2017; Zhang et al., 2013) or considered it in a simple way; e.g., Gao et al. (2017), Huang et al. (2012) and Peng et al. (2016) described the monthly distribution of the total regional emissions according to Moderate-resolution Imaging Spectroradiometer (MODIS) fire count. Although the fire count is a good surrogate for the temporal distribution of open-field burning, its use may have many disadvantages such as a limited detection period, great uncertainty due to cloudy weather, and the inability to include household burning. In addition, the fire count cannot distinguish the type of residue,

which is related to the species of emitted pollutants. Peng et al. (2016) and Wang and Zhang (2008) described the monthly distributions of southern and northern emissions according to the results of questionnaire surveys. Although questionnaire surveys are relatively reliable, there are uncertainties relating to the scope of the survey area and the number of questionnaires. For these reasons, alternative ways of describing the temporal variations are needed. Basing on agricultural practice (e.g., the timings of sowing and harvest) is a possible choice.

Emissions factors (EFs) are fundamentally important in estimating pollutant emissions. Owing to the complexity of the burning conditions and limited on-site experiments, EFs on the whole have large uncertainty (e.g., Akagi et al., 2011; Shen et al., 2014). Previous studies on pollutant emissions from crop residue burning were generally based on EFs obtained in very early and/or limited experiments (Liu et al., 2015; Tian et al., 2011; Venkataraman et al., 2006; Zhang et al., 2016b). In recent years, many experiments for determining EFs of particulates and trace gases from crop straw burning in China have

been conducted (e.g., Tang et al., 2014; Wei et al., 2014; Zhang et al., 2017; Zhang et al., 2013), however, the results have not been well adopted by existing studies. Updating crop-specific EFs with the results of newly conducted experiments is therefore important to improve the emission inventory.

This study presents a comprehensive and high-resolution inventory of current pollutant emissions from crop residue burning in China, including the emissions of BC, OC, $PM_{2.5}$, $PM_{10}$, $SO_2$, $NO_X$, $NH_3$, $CH_4$, NMVOC, CO and $CO_2$. Compared with



previous studies of the same kind, our work has placed more emphasis on spatial and temporal variations. Statistical data at the municipal level combined with high-resolution land use data were adopted to improve the spatial distribution. Detailed crop rotations and harvest times in different regions were considered in determining the temporal distribution. MODIS Thermal Anomalies/Fire products were applied to verify the spatial and temporal variations of the emissions. In addition,

EFs were updated according to the available results of experiments. This study improves our knowledge of the pollutant emissions from crop residue burning in China, and lays an important basis for further research on environmental and health effects.

## 2 Methodology and data

### 2.1 Emission estimation

The present study considered 12 kinds of crop residue namely straws of rice, wheat, corn, barley, millet, sorghum, legume, tuber crop, oil plant, cotton, sugarcane and fiber crop. For simplicity, considering that the major Chinese crops are rice, wheat and corn, the nine remaining less important crops are merged and referred to as "others". In this study, pollutant emissions were firstly estimated for each prefecture-level city according to the yields of crops multiplied by a range of factors, and then redistributed with high-resolution land use data and time coefficients for spatial and temporal variations.

The yearly emissions in a municipal city are calculated as:

$$E_{i,j} = \sum_{n=1}^{n}(C_{i,n} \times R_{i,n} \times B_i \times F \times E_{n,j}), \tag{1}$$

where the subscripts i, j and n respectively stand for a certain city, pollutant species and crop type; $E_{i,j}$ is the total yearly emission of pollutant j in city i (kg yr$^{-1}$); $C_{i,n}$ is the annual yield of crop n in city i (kg yr$^{-1}$); $R_{i,j}$ is the straw-grain ratio (SGR) of crop n in city i (%); $B_i$ is the proportion of crop residue burned in city i (%); F is the burning efficiency (%); $EF_{n,j}$ is the

20 emission factor of pollutant j from the residue burning of crop n (g kg$^{-1}$).

For each type of crop, the production of its residue within a city is obtained by multiplying the yield of the crop by the SGR. The yields of crops at city level ($C_{i,n}$) are published every year by the National Bureau of Statistics of China (NBSC, 2015). The SGRs for most crops were based on a research report entitled "The Crop Straw and Grain Ratio of Main Crops in Different Agricultural Regions", published by the National Development and Reform Commission of China (NDRC, 2015).

The SGRs for sugarcane and fiber crops were based on the findings of Lu et al. (2011) and Zhang et al. (2013). SGRs for different crops used in this study are listed in Table S1. The proportions of the burned crop residues ($B_i$) differ greatly from region to region. They were determined in this study according to the utilization rates of crop residues in different provinces reported by the NDRC (2012), which are listed in Table S2. The burning efficiency is related to many factors, such as the kind of crop straw and burning conditions. De Zárate et al. (2005) used an efficiency of 80 % for total cereal residues. Zhang

et al. (2013) used an efficiency of 90 % for summer rice straw, 95 % for fall rice straw and 95 % for sugarcane leaves. Zhou et al. (2017) assigned efficiencies of 93 %, 92 %, 92 %, 82 %, 90 % and 68 % for rice, wheat, corn, peanut, cotton and





sugarcane, respectively. According to the above studies, we simply assumed the burning efficiency to be 90 % for all kinds of crop residues.

To update the EFs, an in-depth survey of the literature was conducted for studies involving the EFs of crop residue burning. Detailed information of the reported EFs and associated references are presented in Table S3. According to nearly 30 studies conducted in China and abroad, the EFs of eleven pollutants for seven kinds of crop straw were obtained by averaging corresponding values reported in the literature, ignoring obviously unreasonable values. Values of EFs used in this study are listed Table 1. It is noted that crop residues are burned not only in open fields but also as household fuels. Because most experimental studies have focused on field burning, and the reporting of EFs for household burning has been limited, differences in EFs between the two burning modes were not considered in this study.

## 2.2 Spatial and temporal distribution

As mentioned above, spatial distributions were firstly determined on the city level and then redistributed within a city using land use data. For this purpose, an approach based on a geographical information system (GIS) was adopted to perform the redistribution for 296 prefecture-level cities. Raster land use data for 2010 with resolution of 1 km were obtained from the Data Center for Resources and Environmental Sciences, Chinese Academy of Sciences (RESDC) (http://www.resdc.cn). At present, it is difficult to know exactly the ratio of crop residue burned in the open field to that burned as household fuel. According to limited investigations (Cao et al., 2006; Streets et al., 2001; Zhang et al., 2008; Zhou et al., 2017), we assumed the ratio to be 50 %.

The temporal distribution of emissions was determined according to the harvest time of different crops for each of six agricultural regions in China. Information of the harvest time was mainly based on the results of investigations conducted by Wei et al. (2012) and Zhou et al. (2011), which are presented in Table S4. Open-field burning was assumed to happen throughout the crop harvesting period, while household burning was assumed to be equally distributed throughout the year. The temporal distribution was finally expressed in terms of monthly coefficients for each of the main crops in each agricultural region. Monthly emissions could then be obtained by multiplying the yearly emissions by corresponding time coefficients.

## 2.3 Verification with remote sensing data

It is hard to rigorously examine the accuracy of the emission inventory, yet spatial and temporal distributions can be verified by satellite remote sensing data. In this study, data of the MODIS Thermal Anomalies/Fire Daily L3 Global Product (MOD/MYD14A1) for 2014, with 1-km spatial and monthly temporal resolution, was applied (Shon, 2015). The dataset provides the number of burning days for each of the grids (also called pixels, where one pixel has an area of 1 km$^2$) during a period (e.g., a month or a year) for a fire event. To obtain the spatial distribution, the burning intensity for each city was defined as Eq. (2):



$$S_i = \sum_{n=1}^{N} \frac{D_{i,n}}{A_i} \qquad (2)$$

where $S_i$ is the open burning intensity of city $i$ and $D_{i,n}$ is the number of burning days in pixel $n$ of city $i$ within a month or a year. N is the total number of pixels in city $i$ and $A_i$ is the total area of city $i$. Because only agricultural fires were considered in this study, fires on non-farmland were eliminated using the land use data. To verify the temporal distributions, monthly percentages of the fire count were calculated over southern and northern China as well as the whole country.

## 3 Results and discussion

### 3.1 Total pollutant emissions from crop residue burnings in China

The total emissions of 11 pollutants from different crop residue burnings in China in 2014 are shown in Fig. 1. Particulate pollutants include $PM_{10}$, $PM_{2.5}$, OC and BC, with total emissions of 2.66, 2.30, 0.82 and 0.16 Tg, respectively. $PM_{2.5}$ accounts for about 86 % of $PM_{10}$, and most OC and BC is included in $PM_{2.5}$. Particulate pollutants emitted by crop residue burnings are therefore mainly in the form of fine PM. The emission of gaseous pollutants is by far the highest for $CO_2$ (309.04 Tg), followed by CO (13.70 Tg) and NMVOC (1.70 Tg). The next highest emissions are those of $CH_4$ (0.81 Tg) and $NO_X$ (0.70 Tg). Emissions of $NH_3$ and $SO_2$ are relatively low (i.e., 0.14 and 0.09 Tg, respectively). For the total emission of each pollutant, proportions of different crops are similar. The emissions of corn, rice and wheat contribute about 80 % to the total emissions for most pollutants. The contribution is greatest for corn (26 %–46 %), followed by rice (22 %–41 %) and wheat (9 %–18 %). The total emissions came from both open-field burning and household burning, and they are almost equal under the simple assumptions adopted in this study.

To obtain the relative contributions of crop residue burnings to the total anthropogenic emissions of air pollutants, our results were compared with those of the Multiscale Emission Inventory (MEIC) in China (Li et al., 2017b). For particulate pollutants of OC, $PM_{2.5}$, $PM_{10}$ and BC, crop residue burning accounted for 24 %, 19 %, 16 % and 9 % of the total anthropogenic emissions, respectively. For gas pollutants of CO, NMVOC, $CO_2$ and $NO_X$, crop residue burning accounted for 8 %, 7 %, 3 % and 2 %, respectively. For $NH_3$ and $SO_2$, contributions were relatively small.

We also compared our results with those of some previous studies. Cao et al. (2006), for example, estimated emissions of BC (0.29 Tg) and OC (1.17 Tg), and Zhang et al. (2008) estimated emissions of CO (22.59 Tg) and $CO_2$ (252.92 Tg) from both open-filed and household crop residue burnings. Our results for BC (0.16 Tg), OC (0.82 Tg) and CO (13.70 Tg) are lower than results obtained in the two previous studies, yet the emission of $CO_2$ (309.04 Tg) is higher. Wang and Zhang (2008) and Peng et al. (2016) estimated the emissions of the same 10 pollutants that we calculated in our study, but considered only open-field burning. The former study was for the year 2003 while the later was for the year 2009. Wang and Zhang (2008) estimated the emissions of BC, OC, $PM_{2.5}$, $SO_2$, $NO_X$, $NH_3$, $CH_4$, NMVOC, CO and $CO_2$ from open-field straw burning to be 0.05, 0.48, 2.17, 0.06, 0.36, 0.08, 0.37, 0.87, 7.31 and 154.50 Tg, respectively. Peng et al. (2016) estimated the emissions of the same 10 pollutants from open straw burning to be 0.06, 0.41, 1.39, 0.09, 0.42, 0.08, 0.44, 0.94, 5.95 and 143.55 Tg,



respectively. Comparing with our results, we see that except for BC, OC, $PM_{2.5}$, $SO_2$, $NO_X$ and NMVOC emissions in our study are lower than those in the two previous studies and $NH_3$, $CH_4$, CO and $CO_2$ emissions are similar to those in the two previous studies. The generally decreasing trend of the emissions mainly results from the above-mentioned studies being conducted for a base year almost 10 years ago, while open-field burnings in China have remarkably decreased in the past 10 years. The higher emission of BC is mainly due to the larger EF adopted in this study, which was based on the results of newly conducted experiments and may have been underestimated in previous studies.

## 3.2 Spatial distribution of emissions in China

In this study, pollutant emissions from crop residue burning in 2014 were basically calculated for each prefecture-level city in China. Detailed emissions of the 11 pollutants for 296 municipal cities are presented in Table S5. Owing to the space limitation of this article, provincial emissions of major pollutants are summarized in Table 2. Considering that BC has been the most popular species in previous studies because of its important effect on the radiation forcing of the atmosphere, special attention is given to BC in the following discussion of the distribution features. In fact, as shown in Table 2, provincial distributions of the 11 pollutants are similar overall, and their distribution features can generally be reflected by the BC. Figure 2 shows the crop-specific emissions of BC in different provinces, in descending order. As shown, provinces in northeastern (esp., Heilongjiang, Jilin, Liaoning), in northern China (esp., Henan, Shandong) and along the middle and lower reaches of the Yangtze River (esp., Anhui, Hunan) generally have high emissions while provinces in southern and western China generally have lower emissions. In the northeastern China, the burning of corn residues was the major contributor of BC emission. In Jilin, Liaoning and Heilongjiang, for example, corn residue burning accounted for 89 %, 84 % and 81 % of BC emissions, respectively. In northern China, wheat residue burning was generally the major contributor. In Henan, Shandong and Hebei, for example, wheat residue burning accounted for 44 %, 36 % and 31 % of BC emissions, respectively. Along the middle and lower reaches of the Yangtze River, rice residue burning was the major contributor. In Jiangxi, Zhejiang and Hunan, for example, rice residue burning accounted for 81 %, 75 % and 72 % of BC emissions, respectively. These results are consistent with crop distributions in China.

To obtain an emission inventory with finer resolution, the emissions were redistributed within a city using the 1-km resolution land use data. Additionally, we take BC as a representative pollutant for the following discussion. Distributions of BC emission intensity (flux) at 10-km resolution from the residue burning of rice, wheat, corn and all crops for the year 2014 in China are shown in Fig. 3. Fluxes of the total emission (Fig. 3a) were usually high in the northeastern and northern China, along the middle and lower reaches of the Yangtze River, and in the southeastern China, with there being many areas with fluxes higher than 200 kg $km^{-2}$. The provinces with the top five highest fluxes averaged over cropland were Hunan, Anhui, Jilin, Henan and Shandong, with their average fluxes being 103.8, 88.4, 67.5, 57.5 and 55.5 kg $km^{-2}$, respectively. As shown in Fig. 3(b), Fig. 3(c) and Fig. 3(d), the distributions of BC fluxes of corn, wheat and rice are different. Highest fluxes of corn are generally located in areas of northeastern China, such as Jilin, Liaoning, and Heilongjiang. High fluxes of wheat are



generally located in areas of middle-eastern China, such as Anhui, Jiangsu, Henan, and Shandong provinces. High fluxes of rice are generally located in areas of southeastern China, such as Hunan, Sichuan, Anhui, and Hubei provinces.

To verify the spatial distribution with the MODIS fire count, the fire intensities (as defined by Eq. (2)) in each month and throughout the year of 2014 were calculated for each municipal city. Figure 4(a) shows the distribution of the yearly fire intensity, and Fig. 4(b) and Fig. 4(c) respectively show the monthly intensities for June and October, representing the major harvesting seasons for most crops in China. The overall distribution patterns in Fig. 3(a) and Fig. 4(a) agree well. However, there are still obvious disagreements in some places. High intensities of fires are generally located in middle-eastern China in June and in northeastern China in October. This is in consistent with the harvest seasons of major crops in these regions, which will be discussed in more detail in the following section.

## 3.3 Temporal distribution of emissions in China

According to the method introduced in Sect. 2.2, the monthly temporal coefficients for each of the six regions were determined as presented in Table S6. The monthly distributions of the BC emissions over the north, south and whole of China are summarized in Fig. 5. Northern and southern China are divided by a line along the Qinling Mountains and Huaihe River. The two sides obviously differ in terms of climate as and agricultural practice. As shown in Fig. 5, for the whole of China, the temporal distribution of BC emissions over the year peaks in June and October, reflecting the early-summer and mid-autumn as the two major harvest seasons in China. For northern and southern China, although the two sides have similar peaks in June and October, their peak emission intensities are different. The emission in mid-autumn is much higher than that in early-summer in northern China while the emission is higher in early-summer than in mid-autumn in southern China. Additionally, the difference between the two peaks in the south is obviously less than that in the north. This result indicates that pollutant emissions from crop residue burnings are much more serious in northern China, especially during mid-autumn.

The temporal distribution of the BC emissions was also verified with the MODIS fire count. Figure 6 compares monthly proportions between the BC emissions and fire counts of the year. Only open-field burnings are considered in the figure because they are the only burnings that can be seen by satellite. The figure shows that the general patterns of the temporal distributions of BC emissions and fire counts agree well with each other, both having two peaks in June and October. However, for the fire counts obtained by satellite, the peak in June is much higher than that in October, which is not in consistent with the BC emissions. The main reason may be that, fire events are large in number but most small in size in June while they are smaller in number but larger in size in October. This suggests that fire count data from satellites are particularly useful in determining whether burning happen, but less useful in quantifying the emissions.

## 3.4 Uncertainty

Uncertainty analysis is helpful in understanding the results and improving the quality of the inventory. In this study, the Monte Carlo method is used to quantify the uncertainty of the pollutant emissions (e.g., Zhao et al., 2011). Employing a method applied in similar studies (e.g., IPCC, 2006; Peng et al., 2016; Zhou et al., 2017), for EFs, the form of the probability



distribution was assumed to be log-normal, and other factors to be normal distributed. Firstly, the coefficient of variation (CV), which is the standard deviation divided by the mean, was determined for each of the main affecting factors according to the data samples or related studies. The EFs are a major factor affecting the uncertainty of the inventory. In this study, the CV of the EF for each of the major pollutants and crops was determined according to data samples, as summarized in Table S7. There are large differences in the ranges of CVs for different pollutants and crops. In the case of BC, for example, the CVs of different crops vary from 19 % to 26 %, while for OC, the CVs of different crops vary from 15 % to 73 %. The CV for crop yields was assumed to be 10 % following Cao et al. (2008b), while the CV for SGR was assumed to be 5 % because its uncertainty is relatively low. For the proportion of straws to be burned, Wang and Zhang (2008) found the CV to be 20 %–30 % in a questionnaire survey, and we adopted a value of 20 % because the data were based on a more comprehensive study. For the burning efficiency, a CV of 15 % was used following Huang et al. (2012) and related studies.

After determination of the CVs, Monte Carlo simulations were performed using the Crystal Ball (Gonzalez et al., 2004) for 10,000 iterations, and the ranges of emissions (lower and upper percentages relative to the estimated values) with a 95 % confidence interval (CI) were estimated according to simulation results. Table 3 lists the ranges of emissions for the major pollutants and makes comparisons with the results of related studies. As shown in the table, the ranges of emissions are large for different crops, however, for total emissions, the ranges are relatively small, generally falling between −30 % and 60 %. Uncertainty for $CO_2$ is relatively low, at −28 %–35 %, while uncertainty for $NO_X$ is relatively high, at −38 %–64 %. Compared with uncertainties in related studies, uncertainties in our study have been reduced remarkably. This is mainly because the database has been improved in terms of both quality and quantity in the past 10 years. In particular, more experimental results for EFs have become available.

**4 Conclusion**

The total emissions of BC, OC, $PM_{2.5}$, $PM_{10}$, $SO_2$, $NO_X$, $NH_3$, $CH_4$, NMVOC, CO and $CO_2$ from crop residue burning in China in 2014 were estimated to be 0.16, 0.82, 2.30, 2.66, 0.09, 0.70, 0.14, 0.81, 1.70, 13.70 and 309.04 Tg, respectively. Contribution of crop residue burning to the total anthropogenic emissions of air pollutants are large for BC, $PM_{2.5}$ and OC, accounting for about 9 %, 19 %, and 24 %, respectively. Corn, rice and wheat are the three major contributors of emissions, accounting for about 80 % of most pollutants.

The emissions are unevenly distributed spatially and temporally. High emissions are generally located in northeastern China, in northern China, along the middle and lower reaches of the Yangtze River. Even in these areas, the local emission fluxes vary from place to place when high-resolution land use data are applied. Temporally, the emissions generally peak in early-summer and mid-autumn, but are somewhat different in different regions. This implies that crop residue burning may play an important role in air pollution in some regions at some times. In particular, considering also the frequently unfavorable weather conditions in mid-autumn, northern China is most vulnerable to crop residue burnings, and special policies of emission control are needed to alleviate the air pollution.




***Acknowledgments.*** This work was supported by the National Key Research and Development Program of China (2016YFC0208504), and the National Key Basic Research Program of China (2014CB441203).

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

**Table Captions List:**

**Table 1.** Updated EFs of pollutants from crop residue burning.

**Table 2.** Provincial pollutant emissions (Gg) of crop residue burning in 2014.

**Table 3.** Uncertainties of emissions in this study and related studies.

**Figure Captions List:**

**Figure 1.** Pollutant emissions from different crop residue burnings in China in 2014.

**Figure 2.** Provincial BC emissions from different crop residue burnings (Gg) in 2014.

**Figure 3.** Fluxes of pollutant emissions from crop residue burnings in China in 2014: (a) all BC; (b) corn BC; (c) wheat BC; (d) rice BC.

**Figure 4.** Fire intensities of crop residue burnings in China based on MODIS fire counts for (a) the entire year of 2014; (b) June 2014; (c) October 2014.

**Figure 5.** Temporal distributions of BC emissions of crop residue burnings in China in 2014.

**Figure 6.** Comparison of monthly proportions between BC emissions from open straw burning and fire counts by satellite in 2014.





**Table 1.** Updated EFs of pollutants from crop residue burning.

| Pollutants | Emission factors (g kg$^{-1}$) | | | | | |
|---|---|---|---|---|---|---|
| | Rice | Wheat | Corn | Legume | Cotton | Sugarcane |
| BC | 0.58 | 0.68 | 0.75 | 0.63 | 0.82 | 0.73 |
| OC | 3.50 | 3.15 | 3.71 | 1.14 | 3.17 | 1.25 |
| $PM_{2.5}$ | 12.72 | 6.90 | 11.10 | 3.30 | 5.90 | 4.12 |
| $PM_{10}$ | 14.00 | 7.73 | 11.95 | 10.85 | 10.85 | 6.29 |
| $SO_2$ | 0.60 | 0.39 | 0.24 | 0.25 | 0.28 | 0.43 |
| $NO_X$ | 2.96 | 2.28 | 3.36 | 1.56 | 3.03 | 3.49 |
| $NH_3$ | 0.59 | 0.33 | 0.55 | 0.53 | 0.78 | 1.39 |
| $CH_4$ | 4.30 | 3.01 | 3.00 | 3.90 | 3.50 | 4.31 |
| NMVOC | 7.95 | 6.79 | 6.19 | 8.64 | 8.17 | 10.60 |
| CO | 59.20 | 62.20 | 54.50 | 32.90 | 90.05 | 40.08 |
| $CO_2$ | 1355.95 | 1455.38 | 1257.17 | 1445.00 | 1449.20 | 1152.50 |



**Table 2.** Provincial pollutant emissions of crop residue burning in 2014. (Unit for emission estimate: Gg)

| Province | BC | OC | PM$_{2.5}$ | PM$_{10}$ | SO$_2$ | NO$_X$ | NH$_3$ | CH$_4$ | NMVOC | CO | CO$_2$ |
|---|---|---|---|---|---|---|---|---|---|---|---|
| Heilongjiang | 17.3 | 84.6 | 258.4 | 293.2 | 7.1 | 76.7 | 13.6 | 78.7 | 160.9 | 1301.6 | 31126.5 |
| Anhui | 15.2 | 80.0 | 218.9 | 254.8 | 10.1 | 64.7 | 13.1 | 83.7 | 174.8 | 1422.1 | 32256.2 |
| Jilin | 13.3 | 67.3 | 200.7 | 220.6 | 5.1 | 60.3 | 10.5 | 57.8 | 118.7 | 1010.3 | 23241.7 |
| Henan | 12.2 | 63.4 | 150.8 | 178.5 | 6.5 | 50.5 | 10.3 | 58.9 | 129.7 | 1105.3 | 24647.3 |
| Shandong | 11.8 | 58.7 | 149.0 | 168.7 | 5.3 | 48.4 | 9.0 | 52.1 | 114.0 | 1004.3 | 22421.4 |
| Sichuan | 9.3 | 53.9 | 141.0 | 171.2 | 6.4 | 43.9 | 10.4 | 55.1 | 115.6 | 907.4 | 19827.0 |
| Hunan | 8.9 | 53.7 | 168.9 | 195.0 | 7.9 | 44.8 | 10.2 | 61.1 | 119.5 | 903.8 | 19951.8 |
| Hebei | 7.6 | 39.9 | 105.4 | 117.8 | 3.5 | 33.5 | 6.1 | 35.2 | 76.4 | 674.2 | 15085.6 |
| Hubei | 7.6 | 43.9 | 121.1 | 143.7 | 5.6 | 35.8 | 8.4 | 46.5 | 95.9 | 755.2 | 16383.7 |
| Inner Mongolia | 7.4 | 37.2 | 99.9 | 117.0 | 2.8 | 32.7 | 6.4 | 32.5 | 69.7 | 576.7 | 13190.2 |
| Liaoning | 7.3 | 37.5 | 111.5 | 123.4 | 3.0 | 33.4 | 6.0 | 33.0 | 67.6 | 568.9 | 12995.6 |
| Jiangsu | 6.2 | 33.3 | 96.3 | 109.8 | 4.6 | 26.9 | 5.3 | 35.9 | 73.0 | 589.3 | 13431.5 |
| Guangxi | 5.5 | 25.0 | 78.5 | 94.2 | 4.1 | 27.0 | 7.5 | 34.9 | 73.7 | 461.0 | 10897.7 |
| Jiangxi | 3.6 | 21.8 | 70.2 | 81.0 | 3.4 | 18.1 | 4.2 | 25.7 | 49.8 | 372.4 | 8228.6 |
| Gansu | 3.4 | 18.2 | 42.6 | 52.0 | 1.5 | 15.0 | 3.4 | 16.1 | 35.8 | 299.1 | 6489.5 |
| Shanxi | 3.3 | 16.3 | 45.5 | 50.7 | 1.3 | 14.2 | 2.5 | 14.0 | 29.7 | 256.7 | 5917.1 |
| Chongqing | 3.2 | 18.7 | 50.1 | 61.5 | 2.2 | 15.5 | 3.8 | 19.4 | 40.3 | 308.7 | 6743.7 |
| Shananxi | 2.9 | 14.4 | 38.2 | 43.3 | 1.3 | 12.1 | 2.2 | 12.9 | 27.8 | 239.3 | 5432.2 |
| Yunnan | 2.6 | 12.9 | 35.5 | 42.5 | 1.3 | 11.9 | 2.7 | 13.2 | 28.2 | 211.5 | 4868.1 |
| Guangdong | 1.8 | 10.4 | 29.7 | 36.4 | 1.6 | 9.4 | 2.6 | 12.6 | 26.1 | 182.3 | 4007.6 |
| Guizhou | 1.7 | 10.2 | 24.9 | 31.0 | 1.1 | 8.4 | 2.2 | 10.0 | 21.5 | 168.5 | 3561.2 |
| Zhejiang | 1.3 | 7.6 | 24.5 | 28.7 | 1.2 | 6.4 | 1.5 | 9.1 | 17.8 | 130.8 | 2975.2 |
| Ningxia | 0.7 | 4.4 | 9.6 | 12.3 | 0.4 | 3.6 | 0.9 | 3.9 | 8.7 | 70.5 | 1457.3 |
| Fujian | 0.6 | 3.9 | 10.9 | 13.4 | 0.6 | 3.2 | 0.8 | 4.4 | 8.9 | 66.2 | 1423.0 |
| Tianjin | 0.5 | 2.6 | 7.2 | 7.8 | 0.2 | 2.2 | 0.4 | 2.2 | 4.7 | 42.9 | 967.8 |
| Xinjiang | 0.2 | 1.0 | 2.5 | 2.8 | 0.1 | 0.8 | 0.2 | 0.9 | 2.2 | 20.2 | 374.3 |
| Qinghai | 0.2 | 1.2 | 1.7 | 2.7 | 0.1 | 0.9 | 0.3 | 1.1 | 2.6 | 20.6 | 391.7 |
| Shanghai | 0.1 | 0.6 | 2.0 | 2.2 | 0.1 | 0.5 | 0.1 | 0.7 | 1.3 | 9.5 | 216.3 |
| Beijing | 0.1 | 0.4 | 1.3 | 1.4 | 0.0 | 0.4 | 0.1 | 0.4 | 0.8 | 6.9 | 160.4 |
| Xizang | 0.1 | 0.4 | 0.9 | 1.1 | 0.0 | 0.3 | 0.1 | 0.4 | 1.0 | 8.2 | 180.6 |
| Hainan | 0.1 | 0.5 | 1.5 | 1.8 | 0.1 | 0.4 | 0.1 | 0.6 | 1.2 | 8.8 | 189.3 |



**Table 3.** Uncertainties of emissions in this study and related studies.

| Studies and Items | | Range of pollutant emission with 95% CI (%) | | | | |
|---|---|---|---|---|---|---|
| | | BC | NOₓ | CO | CO₂ | NMVOC |
| This study | Emissions by Rice | −55–105 | −58–111 | −57–107 | −53–91 | −52–88 |
| | Emissions by Wheat | −55–105 | −63–160 | −49–68 | −49–68 | −61–139 |
| | Emissions by Corn | −52–92 | −61–147 | −55–101 | −48–70 | −64–161 |
| | Emissions by Others | −54–97 | −57–110 | −54–94 | −51–74 | −58–117 |
| | Total emissions | −33–47 | −38–64 | −32–44 | −28–35 | −36–60 |
| Related studies | Peng et al. (2016)[a] | −78–147 | −49–78 | −91–155 | – | −67–94 |
| | Ni et al. (2015)[b] | – | – | ±148 | ±87 | – |
| | Zhao et al. (2011)[c] | −62–304 | −56–123 | – | – | – |
| | Cao et al. (2008b)[d] | ±59.4 | ±41.7 | ±73.1 | ±59.4 | ±197 |
| | Wang and Zhang (2008)[e] | ±132 | ±80 | ±86 | ±60 | ±71 |

[a, b, d, e] Only open field burnings of crop residues in China were considered.

10 [c] Total biomass burnings in China were considered.

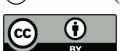



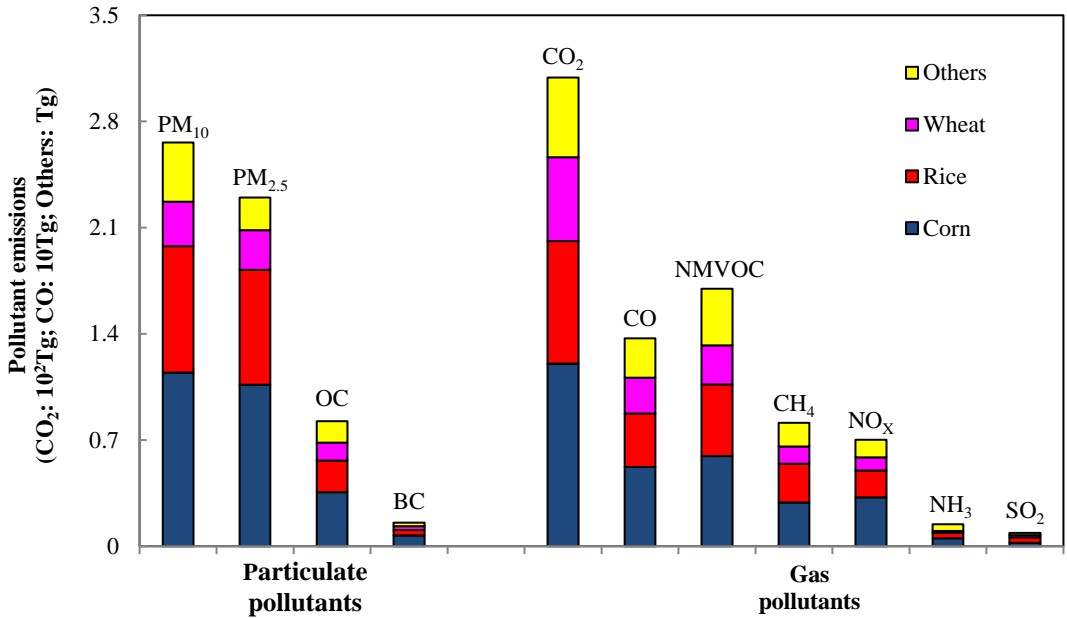

**Figure 1.** Pollutant emissions from different crop residue burnings in China in 2014.

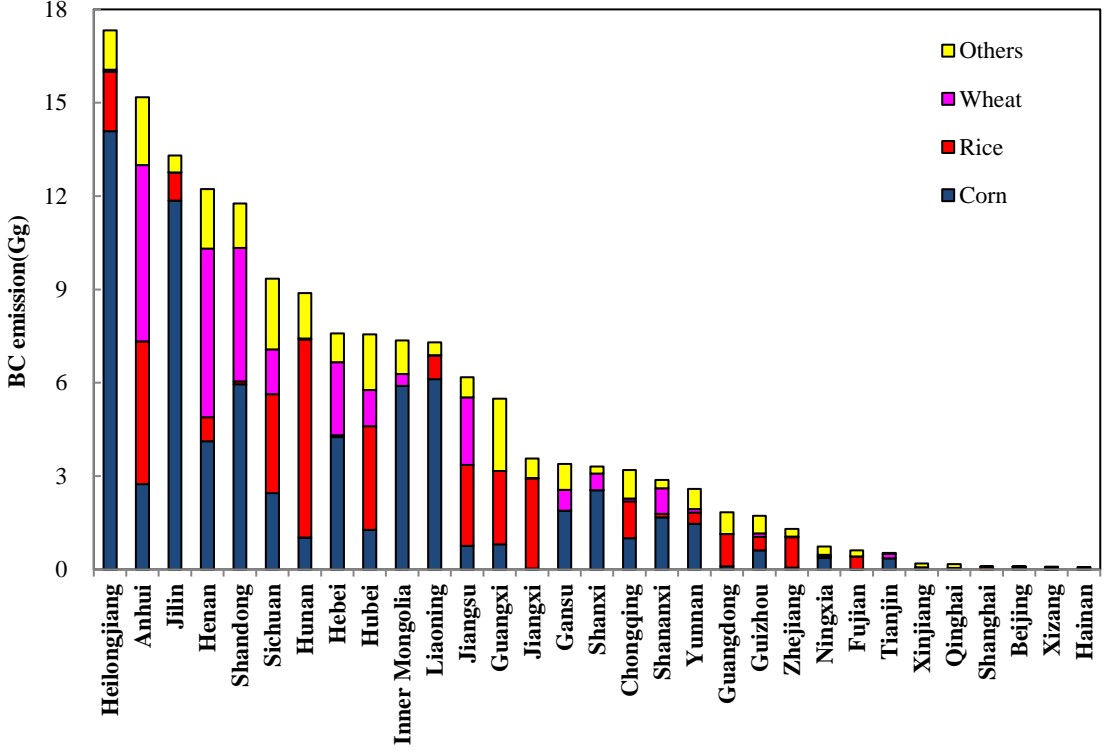

5   **Figure 2.** Provincial BC emissions from different crop residue burnings in 2014. (Unit for emission estimate: Gg)





10    **Figure 3.** Fluxes of pollutant emissions from crop residue burnings in China in 2014: (a) all BC; (b) corn BC; (c) wheat BC; (d) rice BC.



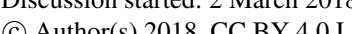

**Figure 4.** Fire intensities of crop residue burnings in China based on MODIS fire counts for (a) the entire year of 2014; (b) June 2014; (c) October 2014.





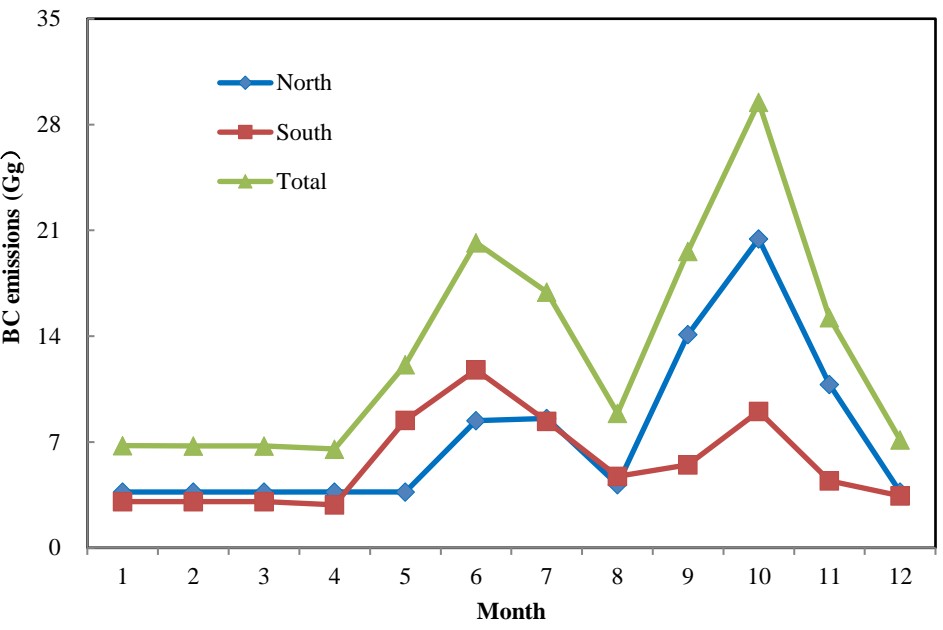

**Figure 5.** Temporal distributions of BC emissions of crop residue burnings in China in 2014.

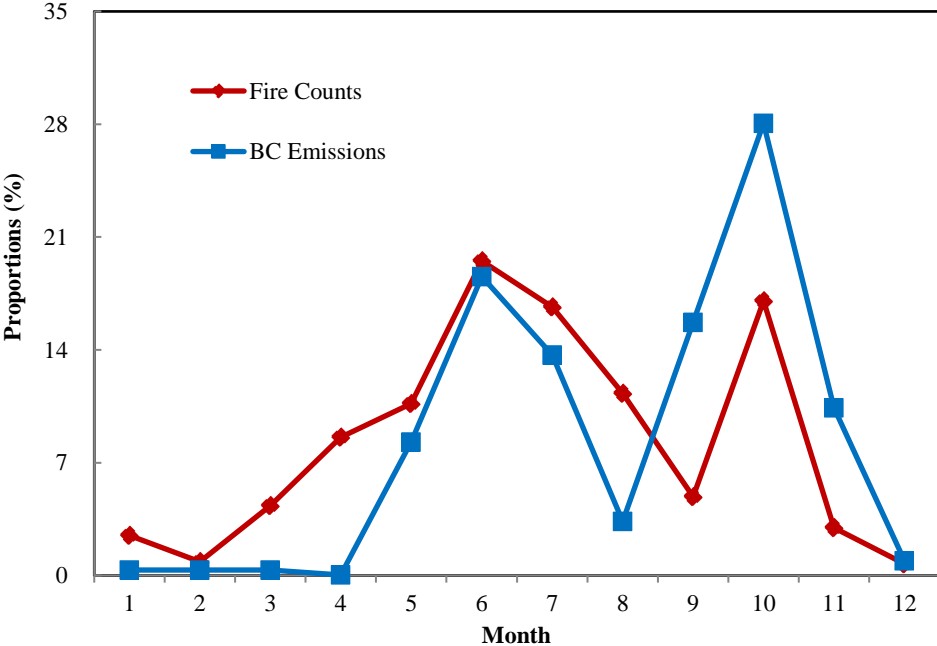

5   **Figure 6.** Comparison of monthly proportions between BC emissions from open straw burning and fire counts by satellite in 2014.