# Peer review of "A high-resolution inventory of air pollutant emissions from crop residue burning in China"

_Atmospheric Chemistry and Physics, 2017_

## Referee Comment (RC1) · Anonymous Referee #1 · 30 Mar 2018

General comments:

Zhang et al. reported an emission inventory of crop residue burning based on the annual yield of crop at city level published in yearbook and the related parameters (e.g., EF, proportion of crop residue burned) without distinction in domestic and in-field burning. The monthly and 1-km spatial variation were obtained based on the farming practice in different regions and land use data, respectively. Currently, there are several papers about the biomass burning emission inventory, including domestic burning and in field burning for different species with high temporal (e.g., daily) and spatial resolution (e.g., 1km). From the abstract and conclusion of this study, this is no difference or new finding compared to the previous studies, such as the main contributor of the emission, the temporal and spatial distribution characteristic. The new information presented in
this paper is not obvious and should be described clearly and in detail. Here I placed several publications for the reference. I hope it will helpful for you to improve your work.

Chen, J., C. Li, Z. Ristovski, A. Milic, Y. Gu, M. S. Islam, S. Wang, J. Hao, H. Zhang, C. He, H. Guo, H. Fu, B. Miljevic, L. Morawska, T. Phong, Y. L. A. M. Fat, G. Pereira, A. Ding, X. Huang & U. C. Dumka (2017) A review of biomass burning: Emissions and impacts on air quality, health and climate in China. Science of the Total Environment, 579, 1000-1034. Gao, R., W. Jiang, W. Gao & S. Sun (2017) Emission inventory of crop residue open burning and its high-resolution spatial distribution in 2014 for Shandong province, China. Atmospheric Pollution Research, 8, 545-554. Zhou, Y., X. F. Xing, J. L. Lang, D. S. Chen, S. Y. Cheng, L. Wei, X. Wei & C. Liu (2017) A comprehensive biomass burning emission inventory with high spatial and temporal resolution in China. Atmospheric Chemistry and Physics, 17, 2839-2864. Qiu, X., L. Duan, F. Chai, S. Wang, Q. Yu & S. Wang (2016) Deriving High-Resolution Emission Inventory of Open Biomass Burning in China based on Satellite Observations. Environmental Science & Technology, 50, 11779-11786. Sun, J., H. Peng, J. Chen, X. Wang, M. Wei, W. Li, L. Yang, Q. Zhang, W. Wang & A. Mellouki (2016) An estimation of CO2 emission via agricultural crop residue open field burning in China from 1996 to 2013. Journal of Cleaner Production, 112, 2625-2631. Ni, H., Y. Han, J. Cao, L. W. A. Chen, J. Tian, X. Wang, J. C. Chow, J. G. Watson, Q. Wang, P. Wang, H. Li & R.-J. Huang (2015) Emission characteristics of carbonaceous particles and trace gases from open burning of crop residues in China. Atmospheric Environment, 123, 399-406.

Specific comments:

P1, L11, "Emissions were firstly estimated for each city and then redistributed using 1-km resolution land use data". P2, L9, "However, almost all existing studies focused only on provincial emissions across China, without considering the detailed distribution within a province". Actually, the preliminary resolution of biomass burning emission has been improved into county-level resolution even into grid resolution without spatial distribution (such as the estimation based on fire radiative power data with 1km

resolution). The paper need more serious and comprehensive literature review.

P2, L10, "However, almost all existing studies focused only on provincial emissions across China, without considering the detailed distribution within a province." The emission of crop residue burning were estimated through the satellite remote sensing data such as MODIS data with grid resolution directly in recent studies, such as Qiu et al. (2016).

P2, L13, "A spatial distribution with finer resolution based on more detailed data of agricultural activity and land use is therefore expected." Actually, there is large uncertainty during the emission distribution process using the land use data as the only proxy (He et al., 2015).

P2, L14, "Meanwhile, previous studies either did not consider the temporal variation (e.g., Chen et al., 2015; Zhang et al., 2017; Zhang et al., 2013) or considered it in a simple way; e.g., Gao et al. (2017), Huang et al. (2012) and Peng et al. (2016) described the monthly distribution of the total regional emissions according to Moderate-resolution Imaging Spectroradiometer (MODIS) fire count." Several studies have researched the daily variation of the crop residue burning, such as Huang et al., 2012, Qiu et al., 2016, Zhou et al., 2017.

P3, L1, "Compared with previous studies of the same kind, our work has placed more emphasis on spatial and temporal variations." Currently, there are many studies about the biomass burning emission inventory with highly temporal (e.g., daily resolution) and spatial resolution (e.g., 1km resolution). How to describe your results are more accurate?

P3, L3, "MODIS Thermal Anomalies/Fire products were applied to verify the spatial and temporal variations of the emissions." If you think the MODIS data "have many disadvantages such as a limited detection period, great uncertainty due to cloudy weather, and the inability to include household burning (P2, L18) ", why you choose MODIS data to verify the spatial and temporal variations of the emissions you studied.

P3, L5, "This study improves our knowledge of the pollutant emissions from crop residue burning in China" In fact, few new knowledge could be found in this study compared to previous research. The author should describe it clearly and in detail.

P3, L17, "$E_{i,j}$ is the total yearly emission of pollutant j in city i", "yearly" should be changed to "annual"

P3, L19, "$B_i$ is the proportion of crop residue burned in city i" There is no difference in cities from the table S2. It should be changed into "$B_i$ is the proportion of crop residue burned in province XXX".

P3, L25, "SGRs for different crops used in this study are listed in Table S1." The SCR of different crops is different. How to determine the SCR of the "Others" in Table S2. P3, L27, "They were determined in this study according to the utilization rates of crop residues in different provinces reported by the NDRC (2012), which are listed in Table S2" Current report showed that the EF of the crop residue domestic burning and in-field burning have great difference (e.g. EPD, 2014). The emission estimation should consider the proportion of domestic burning and in-field burning, respectively. Moreover, the EF selection and emission estimation result also should be given separately according to domestic burning and in-field burning. EPD: Guide for compiling atmospheric pollutant emission inventory for biomass burning, Environmental protection Department, available at: http://www.zhb.gov.cn/gkml/hbb/bgg/201501/t20150107_293955.htm, 2014 (in Chinese).

P4, L1, "According to the above studies, we simply assumed the burning efficiency to be 90 % for all kinds of crop residues." Actually, this treatment is inappropriate. There is different parameter for various crops (He et al., 2015). He, M., Wang, X. R., Han, L., Feng, X. Q. and Mao, X.: Emission Inventory of Crop Residues Field Burning and Its Temporal and Spatial Distribution in Sichuan province, Environmental Science, 36, 1208−1216, 2015 (in Chinese).

P4, L5, "It is noted that crop residues are burned not only in open fields but also as household fuels. Because most experimental studies have focused on field burning, and the reporting of EFs for household burning has been limited, differences in EFs between the two burning modes were not considered in this study." As mentioned above, the EF should be considered in domestic burning and in-field burning separately because of the great difference of EF (EPD, 2014). Actually, there are several reports about the EFs for household burning, such as EPD, 2014, Tang et al. (2014) Wei et al. (2008) Li et al. (2016) Wei et al. (2014).

EPD: Guide for compiling atmospheric pollutant emission inventory for biomass burning, Environmental protection Department, available at: http://www.zhb.gov.cn/gkml/hbb/bgg/201501/t20150107_293955.htm, 2014 (in Chinese). Tang, X. B., Huang, C., Lou, S. R., Qiao, L. P., Wang, H. L., Zhou, M., Chen, M. H., Chen, C. H., Wang, Q., Li, G. L., Li, L., Huang, H. Y. and Zhang, G. F.: Emission Factors and PM Chemical Composition Study of Biomass Burning in the Yangtze River Delta Region, Environmental Science, 35, 1623−1632, 2014 (in Chinese). Wei, W., Wang, S. X., Chatani, S., Klimont, Z., Cofala, J. and Hao, J. M.: Emission and speciation of non−methane volatile organic compounds from anthropogenic sources in China, Atmos. Environ., 42, 4976−4988, doi: 10.1016/j.atmosenv.2008.02.044, 2008. Li, Q., Jiang, J. K., Cai, S. Y., Zhou, W., Wang, S. X., Duan, L. and Hao, J. M.: Gaseous Ammonia Emissions from Coal and Biomass Combustion in Household Stoves with Different Combustion Efficiencies, Environ. Sci. Technol., 3, 98−103, 2016. Wei, S., G. Shen, Y. Zhang, M. Xue, H. Xie, P. Lin, Y. Chen, X. Wang & S. Tao (2014) Field measurement on the emissions of PM, OC, EC and PAHs from indoor crop straw burning in rural China. Environmental Pollution, 184, 18-24.

P4, L11, "As mentioned above, spatial distributions were firstly determined on the city level and then redistributed within a city using land use data. For this purpose, an approach based on a geographical information system (GIS) was adopted to perform the redistribution for 296 prefecture-level cities." As mentioned above, there is large

uncertainty during the emission distribution process using the land use data as the only proxy (He et al., 2015)

P4, L14, "At present, it is difficult to know exactly the ratio of crop residue burned in the open field to that burned as household fuel. According to limited investigations (Cao et al., 2006; Streets et al., 2001; Zhang et al., 2008; Zhou et al., 2016), we assumed the ratio to be 50 %." The ratio of crop residue domestic burning and in-field burning in different regions is very important parameters to estimate the biomass burning emission accurately. This treatment is very inappropriate.

P4, L18, "The temporal distribution of emissions was determined according to the harvest time of different crops for each of six agricultural regions in China." This is a helpful treatment to determine the monthly variation of the emission. However, current studies (e.g., Qiu et al., 2016) could provide daily variation of the emission based on the MODIS data. If the author think the MODIS data will miss several data (P2, L18), why use it for verification (section 2.3)?

---

## Author Comment (AC1) · 15 Apr 2018

**Responses to referee #1**

First of all, we would like to express our sincere gratitude to the referee for the thoughtful, valuable and frank comments, which will be of great help to improve our paper. In order to facilitate reading, the referee's comments are listed in italics and followed by our responses one by one. Numbers are added to the specific comments to ease quotation in the responses.

*General comments:*

*Zhang et al. reported an emission inventory of crop residue burning based on the annual yield of crop at city level published in yearbook and the related parameters (e.g., EF, proportion of crop residue burned) without distinction in domestic and in-field burning. The monthly and 1-km spatial variation were obtained based on the farming practice in different regions and land use data, respectively. Currently, there are several papers about the biomass burning emission inventory, including domestic burning and in field burning for different species with high temporal (e.g., daily) and spatial resolution (e.g., 1km). From the abstract and conclusion of this study, this is no difference or new finding compared to the previous studies, such as the main contributor of the emission, the temporal and spatial distribution characteristic. The new information presented in this paper is not obvious and should be described clearly and in detail. Here I placed several publications for the reference. I hope it will helpful for you to improve your work.*

*Chen, J., C. Li, Z. Ristovski, A. Milic, Y. Gu, M. S. Islam, S. Wang, J. Hao, H. Zhang, C. He, H. Guo, H. Fu, B. Miljevic, L. Morawska, T. Phong, Y. L. A. M. Fat, G. Pereira, A. Ding, X. Huang & U. C. Dumka (2017) A review of biomass burning: Emissions and impacts on air quality, health and climate in China. Science of the Total Environment, 579, 1000-1034.*

*Gao, R., W. Jiang, W. Gao & S. Sun (2017) Emission inventory of crop residue open burning and its high-resolution spatial distribution in 2014 for Shandong province, China. Atmospheric Pollution Research, 8, 545-554.*

*Zhou, Y., X. F. Xing, J. L. Lang, D. S. Chen, S. Y. Cheng, L. Wei, X. Wei & C. Liu (2017) A comprehensive biomass burning emission inventory with high spatial and temporal resolution in China. Atmospheric Chemistry and Physics, 17, 2839-2864.*

*Qiu, X., L. Duan, F. Chai, S.Wang, Q. Yu & S. Wang (2016) Deriving High-Resolution Emission Inventory of Open Biomass Burning in China based on Satellite Observations. Environmental Science & Technology, 50, 11779-11786.*

*Sun, J., H. Peng, J. Chen, X. Wang, M. Wei, W. Li,L. Yang, Q. Zhang, W. Wang & A.*

*Mellouki (2016) An estimation of CO2 emission via agricultural crop residue open field burning in China from 1996 to 2013. Journal of Cleaner Production, 112, 2625-2631.*

*Ni, H., Y. Han, J. Cao, L. W. A. Chen, J. Tian, X. Wang, J. C. Chow, J. G. Watson, Q. Wang, P. Wang, H. Li & R.-J. Huang (2015) Emission characteristics of carbonaceous particles and trace gases from open burning of crop residues in China. Atmospheric Environment, 123, 399-406.*

**General responses:**

Indeed, as an important topic, many researchers have focused on the emissions of air pollutants from crop residue burning in China. However, current results still have large uncertainty because of deficiencies associated with basic data, methodology, as well as some effects of related policies, e.g., for crop residue utilization and air pollution control. This makes a further study like ours necessary and meaningful.

Our study aims to present a more updated, reliable and comprehensive understanding about the emissions of crop residue burning in China. Compared with previous studies, obvious differences can be found in many aspects, such as the focus or topic, methodology, data, and the results as well. Taking the recent papers mentioned by the referee for example, Chen et al. (2017) presented a review of biomass burning in China, focusing on emissions and impacts on air quality, health and climate; Gao et al. (2017) gave a detailed emission inventory of crop residue open burning, but only for the Shandong province; Qiu et al. (2016) derived a high-resolution emission inventory of open biomass burning in China, basing on satellite observations; Sun et al. (2016) estimated the $CO_2$ emission via crop residue open field burning in China from 1996 to 2013; Ni et al. (2015) studied the emission characteristics of carbonaceous particles and trace gases from open burning of crop residues in 2008. Zhou et al. (2017) presented a more comprehensive emission inventory of general biomass burning in China, in which crop residue burning was included. Though our work is similar in this regard, differences between the two studies are still notable in many aspects. For example, in Zhou et al. (2017), the proportion of crop residue burned (PCRB) was assumed to be 45%-65% in most provinces, while in our study it was 15%-30%, which is consistent with the suggested value of 20% by the EPD (2014) (http://www.zhb.gov.cn/gkml/hbb/bgg/201501/t20150107_293955.htm, in Chinese), representing the recent development in crop residue utilization in China. Temporal variation in Zhou et al. (2017) was determined by the MODIS fire counts, while in our study, a method depending on farming practice for different crops in different regions was used. In addition, differences can also be found in activity data, emission factors, and other parameters such as straw to grain ratios (SGRs), burning efficiency, and the final emissions in terms of magnitude and distributions.

Generally, our study is one of very limited studies which focused on emissions of multiple pollutants from crop residue burning at the country scale, based on

bottom-up method with detailed city level activity data and region-specific parameters. Since differences in perspective, methodology and data, our study has presented a new estimation of the present emissions, which helps to improve our understanding deeper and more comprehensively.

However, thanks to the valuable comments and suggestions of the referee, we have realized several defects and problems in our paper, such as no distinction for the in-field and domestic burning, still large uncertainty in the EFs, and inappropriate assumption on the ratio of domestic and infield burning. We will seriously consider all the comments and suggestions from the referee as well as other colleagues to improve our manuscript.

***Specific comments & responses*:**

*(1) P1, L11, "Emissions were firstly estimated for each city and then redistributed using 1-km resolution land use data". P2, L9, "However, almost all existing studies focused only on provincial emissions across China, without considering the detailed distribution within a province". Actually, the preliminary resolution of biomass burning emission has been improved into county-level resolution even into grid resolution without spatial distribution (such as the estimation based on fire radiative power data with 1km resolution). The paper need more serious and comprehensive literature review.*

This study adopted a bottom-up method, and thus the spatial resolution was preliminarily determined by the basic activity data. What we wanted to express was that at the country scale, by the bottom-up method instead of remote sensing, very limited studies of the same kind as ours focused on detailed distribution within a province. We will revise the inappropriate expressions in the manuscript basing on more serious and comprehensive literature review.

*(2) P2, L10, "However, almost all existing studies focused only on provincial emissions across China, without considering the detailed distribution within a province." The emission of crop residue burning were estimated through the satellite remote sensing data such as MODIS data with grid resolution directly in recent studies, such as Qiu et al. (2016).*

As mentioned before, our attentions were placed on studies by the bottom-up method at the country scale, ignoring those by remote sensing. We will express it clear basing on further literature review.

*(3) P2, L13, "A spatial distribution with finer resolution based on more detailed data of agricultural activity and land use is therefore expected." Actually, there is large uncertainty during the emission distribution process using the land use data as the only proxy (He et al., 2015).*

Yes, the uncertainty is still large in the distribution process by using the land use as a proxy. In a study for Sichuan province, He et al. (2015) demonstrated a combination of land use and remote sensing data is a feasible way. However, we think some new questions need to be further considered, e.g., how to weight the two proxies and what additional uncertainty induced by the limitations of the remote sensing. For this reason, in our present study, more efforts have been put into investigating activity data at city level and redistributing with more detailed land use data. As a result, the uncertainty should be improved to some extent and acceptable at least at regional scale, but need further improving at local scale. Anyhow, adopting more proxies can be a good way to improve the distribution, and we will consider this in the future.

*(4) P2, L14, "Meanwhile, previous studies either did not consider the temporal variation (e.g., Chen et al., 2015; Zhang et al., 2017; Zhang et al., 2013) or considered it in a simple way; e.g., Gao et al. (2017), Huang et al. (2012) and Peng et al. (2016) described the monthly distribution of the total regional emissions according to Moderate resolution Imaging Spectroradiometer (MODIS) fire count." Several studies have researched the daily variation of the crop residue burning, such as Huang et al., 2012, Qiu et al., 2016, Zhou et al., 2017.*

We have noticed that several studies have researched monthly and even daily variation of crop residue burning, but most of them were based on the MODIS fire counts, and this was what we meant by the "in a simple way" (we have realized this is an inappropriate expression, and will revise it later). In this study, an alternative method was introduced to determine the monthly variation, basing on farming practice in different regions, and the result by remote sensing was used for comparison. Since both methods have their advantages and also disadvantages at least at describing the monthly variation, we think it is meaningful to use different methods and compare their agreement with each other. In addition, we think daily variation may have much larger uncertainty due to its randomness, and in this sense the result may be not as much significant as monthly variation.

*(5) P3, L1, "Compared with previous studies of the same kind, our work has placed more emphasis on spatial and temporal variations." Currently, there are many studies about the biomass burning emission inventory with highly temporal (e.g., daily resolution) and spatial resolution (e.g., 1km resolution). How to describe your results are more accurate?*

In this study, spatial variation was determined preliminarily by activity data at city level and related region-specific parameters (e.g., SGRs and PCRB), and redistributed to finer grids within a city by detailed types of land use data. Temporal variation was determined by related farming practice for different crops in different regions. Basing on a bottom-up method with more detailed data, the reliability and accuracy can be improved, compared with previous studies of the same kind at the country scale. In

addition, the comparison on spatial and temporal variations with the results from a different method basing on remote sensing (MODIS fire counts) is helpful for related studies in future.

*(6) P3, L3, "MODIS Thermal Anomalies/Fire products were applied to verify the spatial and temporal variations of the emissions." If you think the MODIS data "have many disadvantages such as a limited detection period, great uncertainty due to cloudy weather, and the inability to include household burning (P2, L18) ", why you choose MODIS data to verify the spatial and temporal variations of the emissions you studied.*

In this study, the spatial and temporal variations from bottom-up method were compared with those from remote sensing. Since both methods have their advantages and disadvantages, we think it might be useful to compare their agreement with each other. However, the "verify" may be an inappropriate expression. We will describe this purpose in more detail and revise related expressions in revised manuscript.

*(7) P3, L5, "This study improves our knowledge of the pollutant emissions from crop residue burning in China" In fact, few new knowledge could be found in this study compared to previous research. The author should describe it clearly and in detail.*

As mentioned in the general responses, the most remarkable features or new knowledge of our study can be described as follow:
- Generally, this study is one of very limited studies which focused on emissions of multiple pollutants from crop residue burning at the country scale. Basing on a bottom-up oriented method with more detailed, updated and regionally specific data, this study has presented a new and more comprehensive estimation of the emissions, compared with previous studies of the same kind.
- Spatial variation was determined preliminarily by first-hand activity data at city level and related regional parameters (e.g., SGRs and PCRB), and redistributed to finer grids within a city by detailed types of land use data. As a result, the reliability and accuracy of the spatial distribution can be generally improved.
- Temporal variation was determined by farming practice for different crops in different regions. This is quite different from the remote sensing based method which has been widely used in current studies. It provides a new viewpoint of the temporal variation, and we believe the result has advantages at least for the monthly variation, considering the limitations of the remote sensing data.
- The comparison between the results by the bottom-up method and the remote sensing, respectively, may be useful for related studies in the future.

*(8) P3, L17, "$E_{i,j}$ is the total yearly emission of pollutant j in city i", "yearly" should be changed to "annual"*

We will correct it according to this suggestion.

*(9) P3, L19, "Bi is the proportion of crop residue burned in city i" There is no difference in cities from the table S2. It should be changed into "Bi is the proportion of crop residue burned in province XXX".*

We will correct it according to this suggestion.

*(10) P3, L25, "SGRs for different crops used in this study are listed in Table S1." The SGR of different crops is different. How to determine the SGR of the "Others" in Table S1.*

The SCR of the "Others" in Table S1 was assigned with the value also for "others" suggested by the NDRC (2015) (available at website listed in the references)

*(11) P3, L27, "They were determined in this study according to the utilization rates of crop residues in different provinces reported by the NDRC (2012), which are listed in Table S2" Current report showed that the EF of the crop residue domestic burning and in-field burning have great difference (e.g. EPD, 2014). The emission estimation should consider the proportion of domestic burning and in-field burning, respectively. Moreover, the EF selection and emission estimation result also should be given separately according to domestic burning and in-field burning. EPD: Guide for compiling atmospheric pollutant emission inventory for biomass burning, Environmental protection Department, available at: http://www.zhb.gov.cn/ gkml/ hbb/ bgg/ 201501/t20150107_293955.htm, 2014 (in Chinese).*

We appreciate the referee very much for this comment. We have realized that the distinction between in-field and domestic burning is necessary, and will make more serious and comprehensive literature review to distinguish the EFs and other related parameters, separated for in-field and domestic burning.

*(12) P4, L1, "According to the above studies, we simply assumed the burning efficiency to be 90 % for all kinds of crop residues." Actually, this treatment is inappropriate. There is different parameter for various crops (He et al., 2015). He, M., Wang, X. R., Han, L., Feng, X. Q. and Mao, X.: Emission Inventory of Crop Residues Field Burning and Its Temporal and Spatial Distribution in Sichuan province, Environmental Science, 36, 1208-1216, 2015 (in Chinese).*

The burning efficiency depends on many factors such as type of crops, weather, and way of burning, but the reported values in related studies (e.g., He et al., 2015; Tian et al., 2011; Zhang et al., 2013; Gao et al., 2017) are not different so much, mostly from 80% to 95%. For this reason, we adopted a uniform value of 90% for different crops, which is also suggested by the EPD (2014). We will try to make distinction in major crops and regions after further literature review.

*(13) P4, L5, "It is noted that crop residues are burned not only in open fields but also as household fuels. Because most experimental studies have focused on field burning, and the reporting of EFs for household burning has been limited, differences in EFs between the two burning modes were not considered in this study." As mentioned above, the EF should be considered in domestic burning and in-field burning separately because of the great difference of EF (EPD, 2014). Actually, there are several reports about the EFs for household burning, such as EPD, 2014, Tang et al. (2014) Wei et al. (2008) Li et al. (2016) Wei et al. (2014).*

*EPD: Guide for compiling atmospheric pollutant emission inventory for biomass burning, Environmental protection Department, available at: http://www.zhb.gov.cn/gkml/hbb/bgg/201501/t20150107_293955.htm, 2014 (in Chinese). Tang, X. B., Huang, C., Lou, S. R., Qiao, L. P., Wang, H. L., Zhou, M., Chen, M. H., Chen, C. H., Wang, Q., Li, G. L., Li, L., Huang, H. Y. and Zhang, G. F.: Emission Factors and PM Chemical Composition Study of Biomass Burning in the Yangtze River Delta Region, Environmental Science, 35, 1623□1632, 2014 (in Chinese). Wei, W., Wang, S. X., Chatani, S., Klimont, Z., Cofala, J. and Hao, J. M.: Emission and speciation of non-methane volatile organic compounds from anthropogenic sources in China, Atmos. Environ., 42, 4976-4988, doi: 10.1016/j.atmosenv.2008.02.044, 2008. Li, Q., Jiang, J. K., Cai, S. Y., Zhou, W., Wang, S. X., Duan, L. and Hao, J. M.: Gaseous Ammonia Emissions from Coal and Biomass Combustion in Household Stoves with Different Combustion Efficiencies, Environ. Sci. Technol., 3, 98-103, 2016. Wei, S., G. Shen, Y. Zhang, M. Xue, H. Xie, P. Lin, Y. Chen, X. Wang & S. Tao (2014) Field measurement on the emissions of PM, OC, EC and PAHs from indoor crop straw burning in rural China. Environmental Pollution, 184, 18-24.*

Again, we appreciate the referee very much for suggesting more references about the EFs. As mentioned before, we will deal with the domestic and infield burning separately. More efforts will be made to literature review to get a better dataset of the EFs, and we believe the manuscript will be improved obviously in this regard.

*(14) P4, L11, "As mentioned above, spatial distributions were firstly determined on the city level and then redistributed within a city using land use data. For this purpose, an approach based on a geographical information system (GIS) was adopted to perform the redistribution for 296 prefecture-level cities." As mentioned above, there is large uncertainty during the emission distribution process using the land use data as the only proxy (He et al., 2015)*

We think this comment is similar to the (3), please refer to the response there.

*(15) P4, L14, "At present, it is difficult to know exactly the ratio of crop residue burned in the open field to that burned as household fuel. According to limited investigations (Cao et al., 2006; Streets et al., 2001; Zhang et al., 2008; Zhou et al., 2016), we assumed the ratio to be 50 %." The ratio of crop residue domestic burning and infield burning in different regions is very important parameters to estimate the*

*biomass burning emission accurately. This treatment is very inappropriate.*

This will become more important when distinction in the EFs is make for domestic and infield burning. We thank the referee for this comment, and will distinguish the ratio for different regions basing on further literature review.

*(16) P4, L18, "The temporal distribution of emissions was determined according to the harvest time of different crops for each of six agricultural regions in China." This is a helpful treatment to determine the monthly variation of the emission. However, current studies (e.g., Qiu et al., 2016) could provide daily variation of the emission based on the MODIS data. If the author think the MODIS data will miss several data (P2, L18), why use it for verification (section 2.3)?*

We agree that remote sensing method has obvious advantage in describing the temporal variation with high resolution. In this study, the bottom-up oriented method was used, which is essentially different from the remote sensing. As mentioned before, both methods have their advantages and disadvantages at least for monthly variation. For this reason, we think it might be useful to compare their agreement with each other. As suggested by the referee, combination of the two method can be a good way to improve the result. We will focus in this area in future.

**References**

Bi, Y. Y.: Study on Straw Resources Evaluation and Utilization in China, PhD thesis, Chinese Academy of Agriculture Sciences, China, Beijing, 2010 (in Chinese).

Cao, G. L., Zhang, X. Y., and Zheng, F. C.: Inventory of black carbon and organic carbon emissions from China, Atmos. Environ., 40, 6516–6527, doi: 10.1016/j.atmosenv.2006.05.070, 2006.

Chen, G. Y., Guan, Y. N., Tong, L., Yan, B. B., and Hou, L. A.: Spatial estimation of $PM_{2.5}$ emissions from straw open burning in Tianjin from 2001 to 2012, Atmos. Environ., 122, 705–712, doi: 10.1016/j.atmosenv.2015.10.043, 2015.

Chen, J. M., Li, C. L., Ristovski, Z., Milic, A., Gu, Y. T., Islam, M. S., Wang, S. X., Hao, J. M., Zhang, H. F., He, C. R., Guo, H., Fu, H. B., Miljevic, B., Morawska, L., Thai, P., Yun, F. L., Pereira, G., Ding, A. J., Huang, X., Dumka, U. C.: A review of biomass burning: Emissions and impacts on air quality, health and climate in China. Sci. Total Environ., 579, 1000–1034, doi: 10.1016/ j.scitotenv. 2016.11.025, 2017.

EPD: Guide for compiling atmospheric pollutant emission inventory for biomass burning, Environmental Protection Department, available at: http://www. zhb.gov.cn/gkml/hbb/bgg/201501/t20150107_293955.htm, 2014 (in Chinese).

Gao, R., Jiang, W., Gao, W. D., and Sun, S. D.: Emission inventory of crop residue open burning and its high-resolution spatial distribution in 2014 for Shandong province, China, Atmos. Pollut. Res., 8, 545–554, doi: 10.1016/j.apr. 2016.12.009, 2017.

He, M., Wang, X. R., Han, L., Feng, X. Q., and Mao, X.: Emission inventory of crop residues field burning and its temporal and spatial distribution in Sichuan province, Environ. Sci., 36, 1208–1216, 2015 (in Chinese).

Huang, K. H., Fu, J. S., Hsu, N. C., Gao, Y., Dong, X., Tsay, S.-C., and Lam, Y. F.: Impact assessment of biomass burning on air quality in Southeast and East Asia during BASE-ASIA, Atmos. Environ., 78, 291–302, doi: 10.1016/j.atmosenv. 2012.03.048, 2012.

Li, Q., Jiang, J. K., Cai, S. Y., Zhou, W., Wang, S. X., Duan, L., and Hao, J. M.: Gaseous ammonia emissions from coal and biomass combustion in household stoves with different combustion efficiencies, Environ. Sci. Technol., 3, 98–103, 2016.

NDRC (National Development and Reform Commission): The Crop Straw and Grain Ratio of Main Crops form Different Agricultural Regions, Development and Reform Office, available at: http://www.ndrc.gov.cn/zcfb/zcfbtz/201512/ t20151216_767695.html, 2015 (In Chinese).

NDRC (National Development and Reform Commission): National Comprehensive Utilization and Incineration of Straw (In Chinese), Development and Reform Office, available at: http://www.ndrc.gov.cn/zcfb/zcfbtz/201403/ t20140317 _602802.html, 2014 (In Chinese).

Ni, H. Y., Han, Y. M., Cao, J. J., Chen, L. W. A., Tian, J., Wang, X. L., Chow, J. C., Watson, J. G., Wang, Q. Y., Wang, P., Li, H., and Huang, R. J.: Emission characteristics of carbonaceous particles and trace gases from open burning of crop residues in China, Atmos. Environ., 123, 399–406, doi: 10.1016/j.atmosenv. 2015.05.007, 2015.

Peng, L., Zhang, Q., and He, K.: Emissions inventory of atmospheric pollutants from open burning of crop residues in China based on a national questionnaire, Res. Environ. Sci., 29, 1109–1118, doi: 10.13198/j.issn. 1001–6929.2016.08.02, 2016 (in Chinese).

Qiu, X. H., Duan, L., Chai, F. H., Wang, S. X., Yu, Q., Wang, S. L.: Deriving high-resolution emission inventory of open biomass burning in China based on satellite observations. Environ. Sci. Technol., 21, 11779–11786, doi: 10.1021/ acs.est.6b02705, 2016.

Streets, D. G., Gupta, S., Waldhoff, S. T., Wang, M. Q., Bond, T. C., and Bo, Y. Y.: Black carbon emissions in China, Atmos. Environ., 35, 4281–4296, doi: 10.1016/ S1352-2310(01)00179-0, 2001.

Sun, J. F., Peng, H. Y., Chen, J. M., Wang, X. M., Wei, M., Li, W. J., Yang, L. X., Zhang, Q. Z., Wang, W. X., Mellouki, A.: An estimation of $CO_2$ emission via agricultural crop residue open field burning in China from 1996 to 2013. J. Clean Prod., 112, 2625–2631, doi: 10.1016/j.jclepro.2015.09.112, 2016.

Tang, X. B., Huang, C., Rong, L. S., Ping, L. P., Wang, H. L., Zhou, M., Chen, M. H., Chen, C. H., Wang, Q., Li, Q. L., Li, L., Huang, H. Y., and Zhang, G. F.: Emission factor and PM chemical composition study of biomass burning in the Yangtze River Delta Reigon, Environmental Science, 35, 1623–1632, doi: 10.13227/ j.hjkx. 2014.05.001, 2014 (in Chinese).

Tian, H. Z., Zhao, D., and Wang, Y.: Emission inventories of atmospheric pollutants discharged from biomass burning in China, Acta. Sci. Circum., 31, 349–357, doi: 10.13671/j.hjkxxb. 2011.02.018, 2011 (in Chinese).

Wei, S. Y., Shen, G. F., Zhang, Y. Y., Xue, M., Xie, H., Lin, P. C., Chen, Y. C., Wang, X. L., and Tao, S.: Field measurement on the emissions of PM, OC, EC and PAHs from indoor crop straw burning in rural China, Environ. Pollut.,184, 18–24, doi: 10.1016/j.envpol.2013.07.036, 2014.

Wei, W., Wang, S. X., Chatani, S., Klimont, Z., Cofala, J. and Hao, J. M.: Emission and speciation of non-methane volatile organic compounds from anthropogenic sources in China, Atmos. Environ., 42, 4976–4988, doi: 10.1016/j.atmosenv. 2008.02.044, 2008.

Zhang, F. C. and Zhu, Z. H.: Harvest Index of Crops in China, Scientia Agricultura Sinica, 23, 83–87, 1990 (in Chinese).

Zhang, H. F., Hu, J., Qi, Y. X., Li, C. L., Chen, J. M., Wang, X. M., He, J. W., Wang, S. X., Hao, J. M., Zhang, L. L., Zhang, L. J., Zhang, Y. X., Li, R. K., Wang, S. L., and Chai, F. H.: Emission characterization, environmental impact, and control measure of $PM_{2.5}$ emitted from agricultural crop residue burning in China, J. Clean Prod., 149, 629–635, doi: 10.1016/j.jclepro.2017.02.092, 2017.

Zhang, H. F., Ye, X. N., Cheng, T. T., Chen, J. M., Yang, X., Wang, L., and Zhang, R. Y.: A laboratory study of agricultural crop residue combustion in China: Emission factors and emission inventory, Atmos. Environ., 42, 8432–8441, doi: 10.1016/j.atmosenv.2008.08.015, 2008.

Zhang, Y. S., Shao, M., Lin, Y., Luan, S. J., Mao, N., Chen, W. T., and Wang, M.: Emission inventory of carbonaceous pollutants from biomass burning in the Pearl River Delta Region, China, Atmos. Environ., 76, 189–199, doi: 10.1016/ j.atmosenv.2012.05.055, 2013.

Zhou, Y., Xing, X. F., Lang, J. L., Chen, D. S., Cheng, S. Y., Wei, L., Wei, X., and Liu, C.: A comprehensive biomass burning emission inventory with high spatial and temporal resolution in China, Atmos. Chem. Phys., 17, 2839–2864 doi: 10.5194/acp-17-2839-2017, 2017.

---

## Referee Comment (RC2) · Dr. Chen (Referee) · 24 May 2018

Zhang et al. reported an emission inventory of major air pollutants from crop residue burning for the year of 2014. The monthly and 1-km spatial variation were obtained based on the farming practice in 296 prefecture-level cities. The work is interesting, and suitable for the ACP readers. Some important papers should be referred to overview the updated research on this field: 1)Chen, J., C. Li, Z. Ristovski et al., 2017, A review of biomass burning: Emissions and impacts on air quality, health and climate in China. Science of the Total Environment, 579, 1000-1034. 2)Zhou, Y., X. F. Xing, J. L. Lang et al., 2017, A comprehensive biomass burning emission inventory with high spatial and temporal resolution in China. Atmospheric Chemistry and Physics, 17, 2839-2864.

[Figure]

Minor Revision: 1)Line 27, P1, "with most (85 %) being corn, wheat and rice straw" could delete "(85%)"; 2)From Lin33 P2 to Line 7 P3, the paragraph should be shorten. 3)Line 22 P5, about "For NH3 and SO2, contributions were relatively small.", I suggest authors should give the data how much they? As the importance of NH3 and SO2 as precusors for ammonium and sulfate, it should conclude. 4)In table 3, what does it mean, for example, "ïij■55–105"? 5) Figure 2 P16, does it can be divided from regions ïijĹ6 regionsïijĽïij§

---

## Author Comment (AC2) · 25 May 2018

First of all, we would like to thank Dr. Chen for the valuable comments, which will be of great help for improving our paper. In order to facilitate reading, the original comments from the referee are listed in italics and answered one by one.

*General comment and response:*
*Zhang et al. reported an emission inventory of major air pollutants from crop residue burning for the year of 2014. The monthly and 1-km spatial variation were obtained based on the farming practice in 296 prefecture-level cities. The work is interesting, and suitable for the ACP readers. Some important papers should be referred to overview the updated research on this field:*
*Chen, J., C. Li, Z. Ristovski, A. Milic, Y. Gu, M. S. Islam, S. Wang, J. Hao, H. Zhang, C. He, H. Guo, H. Fu, B. Miljevic, L. Morawska, T. Phong, Y. L. A. M. Fat, G. Pereira, A. Ding, X. Huang & U. C. Dumka (2017) A review of biomass burning: Emissions and impacts on air quality, health and climate in China. Science of the Total Environment, 579, 1000-1034.*
*Zhou, Y., X. F. Xing, J. L. Lang, D. S. Chen, S. Y. Cheng, L. Wei, X. Wei & C. Liu (2017) A comprehensive biomass burning emission inventory with high spatial and temporal resolution in China. Atmospheric Chemistry and Physics, 17, 2839-2864.*

We appreciate the referee for the positive comments on our paper and suggestion for additional references. Besides the two papers listed above, more literature review will be made and current research in this field will be updated in revised manuscript.

**Specific comments & responses:**

*(1) Line 27, P1, "with most (85 %) being corn, wheat and rice straw"could delete "(85%)";*
We accept this suggestion and will delete this expression.

*(2) From Lin33 P2 to Line 7 P3, the paragraph should be shorten.*
We will shorten this paragraph, and improve the writing of the whole manuscript.

*(3) Line 22 P5, about "For $NH_3$ and $SO_2$, contributions were relatively small." I suggest authors should give the data how much they? As the importance of $NH_3$ and $SO_2$ as precusors for ammonium and sulfate, it should conclude.*
Yes, the two gases are important precursors of the secondary aerosol. The detailed

data of their contributions will be given in the revised version.

*(4) In table 3, what does it mean, for example, "-55–105"?*
It means the lower and upper limits of the intervals with 95% confidence, from -55%
to 105% of the mean emission. We are regret that it was not described clear in table 3
and will revise it later.

*(5) Figure 2 P16, does it can be divided from regions ïj ´ L6 regionsïjL'ïj§*
Because of the misprints in this comment, we are not quite sure about what the referee
meant exactly. However, we understand that it would be interesting if the provinces
shown in figure 2 were divided into different regions, as mentioned in Line 15-20 P6,
and in table S6 of the supplement. We will consider to do it in the revision of the
manuscript.